# A Piezoelectric Micromachined Ultrasonic Transducer-Based Bone Conduction Microphone System for Enhancing Speech Recognition Accuracy

**DOI:** 10.3390/mi16060613

**Published:** 2025-05-23

**Authors:** Chongbin Liu, Xiangyang Wang, Jianbiao Xiao, Jun Zhou, Guoqiang Wu

**Affiliations:** 1The Institute of Technological Sciences, Wuhan University, Wuhan 430072, China; liuchongbin@whu.edu.cn (C.L.); kent_wxy@whu.edu.cn (X.W.); 2School of Information and Communication Engineering, University of Electronic Science and Technology of China (UESTC), Chengdu 611731, China; jianbiao_x@std.uestc.edu.cn (J.X.); zhouj@uestc.edu.cn (J.Z.); 3Hubei Yangtze Memory Laboratories, Wuhan 430205, China

**Keywords:** microelectromechanical systems (MEMS), piezoelectric micromachinedultrasonic transducer (PMUT), bone conduction microphone, speech enhancement

## Abstract

Speech recognition in noisy environments has long posed a challenge. Air conduction microphone (ACM), the devices typically used, are susceptible to environmental noise. In this work, a customized bone conduction microphone (BCM) system based on a piezoelectric micromachined ultrasonic transducer is developed to capture speech through real-time bone conduction (BC), while a commercial ACM is integrated for simultaneous capture of speech through air conduction (AC). The system enables simpler and more robust BC speech capture. The BC speech capture achieves a signal-to-noise amplitude ratio over five times greater than that of AC speech capture in an environment with a noise level of 68 dB. Instead of using only AC-captured speech, both BC- and AC-captured speech are input into a speech enhancement module. The noise-insensitive BC-captured speech serves as a speech reference to adapt the SE backbone of AC-captured speech. The two types of speech are fused, and noise suppression is applied to generate enhanced speech. Compared with the original noisy speech, the enhanced speech achieves a character error rate reduction of over 20%, approaching the speech recognition accuracy of clean speech. The results indicate that this speech enhancement method based on the fusion of BC- and AC-captured speech efficiently integrates the features of both types of speech, thereby improving speech recognition accuracy in noisy environments. This work presents an innovative system designed to efficiently capture BC speech and enhance speech recognition in noisy environments.

## 1. Introduction

As a crucial component of human–computer interfaces, speech recognition technology has been widely applied in various fields [1,2,3]. However, recognizing speech in noisy environments has long presented a challenge. Noise interference can significantly degrade the quality and intelligibility of speech. In response to this challenge, speech enhancement technology has emerged. Speech enhancement technology refers to the application of various signal processing and machine learning methods to process speech signals, thereby improving their clarity and intelligibility [4]. The primary objective is to improve speech quality in noisy environments, enabling machines to comprehend spoken content more clearly.

Speech enhancement can be categorized into digital signal processing (DSP) methods and deep learning-based methods [4,5]. In recent years, deep learning-based speech enhancement methods have brought significant advancements to this field [6,7,8]. By leveraging the powerful nonlinear mapping capabilities of neural networks, these methods effectively suppress transient non-stationary noise and can significantly reduce background noise from unfamiliar speakers and noise types. However, in the event of an extremely low signal-to-noise ratio (SNR), the aforementioned speech enhancement methods appear somewhat ineffective [9]. This is because noise dominates the sound signal, making it challenging to recover clear speech. These uncontrollable noises arise because traditional speech capture relies on air conduction microphones, which are susceptible to noise and contamination from the surrounding environment. In contrast, bone conduction microphones directly capture vibrations from bones, minimizing the influence of ambient noise and resulting in a higher SNR [10,11]. Consequently, researchers have explored speech enhancement technology based on bone conduction speech [12,13,14,15].

The disadvantage of bone conduction speech is the attenuation of high-frequency signals, which results in reduced audibility compared to air conduction speech [16,17,18,19]. Thus, in early research, bone conduction signals were utilized to extract auxiliary speech information under noisy conditions, such as speech activity detection, SNR estimation, and pitch extraction [20,21]. Subsequently, researchers sought to extend the bandwidth of BC signals to enhance speech quality [22,23]. These methods can be categorized into three types: equalization, analysis and synthesis, and deep neural network (DNN)-based approaches [24,25,26,27,28,29]. Recently, feature fusion algorithms for BC- and AC-captured speech have achieved remarkable performance in speech enhancement [30,31,32,33,34]. This method is increasingly being applied in the fields of speaker recognition and speech recognition.

Furthermore, traditional bone conduction microphones are typically based on accelerometers or modified air conduction microphones [35]. Bone conduction microphones based on accelerometers demonstrate poor performance in continuous speech pickup, while those adapted from air conduction microphones possess complex structures and packaging. At present, fiber-optic sensors also show excellent results in voice detection and recognition, even in the case of a distributed operating mode [36,37].

This work proposes a bone conduction microphone system based on a PMUT to achieve a simpler and more robust device for bone conduction, enabling effective continuous collection of speech by BC. Figure 1 provides an overview of the entire system developed in this work.

PMUTs exhibit a wide bandwidth and high noise resolution at low frequencies and has been applied in the Internet of Medical Things (IoMT) [38]. Compared with MEMS capacitive microphones, PMUTs have advantages such as simple structure, no need to control the capacitor gap, reduced process difficulty and cost, and low power consumption. The speech collected by a BCM and an ACM is transmitted in real time to the application. Following speech enhancement through attentional feature fusion (AFF) [6,39] and deep complex convolutional recurrent network (DCCRN) [40], the speech is decoded online using a trained acoustic model based on deep neural networks (DNNs). Compared to the original noisy speech, the enhanced speech results in higher speech recognition accuracy. This method exploits the noise insensitivity of BC-captured speech and the large bandwidth of AC-captured speech, thereby improving speech recognition accuracy in noisy environments. It can be used in various applications, including speech-based human interaction, far-field speech recognition, and AI technologies.

It is worth noting that the design of this study is centered on effective bone conduction speech acquisition using a PMUT and speech enhancement using the advantages of the respective acoustic features of bone conduction and air conduction speech signals. The AFF-DCCRN (attentional feature fusion + deep complex convolution recurrent network) is a suitable choice of speech enhancement neural network for validation. Although the quality of the speech can be further improved by improving the algorithms, these optimizations are not usually constrained by the hardware architecture.

## 2. Design of Bone Conduction Microphone System

In our previous work, we reported a bone conduction pickup method positioned at the throat [41]. In this work, both the front-end sensor component and the overall mold of the bone conduction microphone system have been upgraded. Additionally, a commercial microphone from STMicroelectronics is integrated into the system. The schematic diagram of the reported PMUT-based bone conduction microphone system is illustrated in Figure 2.

The packaged PMUT, ACM, and amplification circuit are integrated onto a single printed circuit board (PCB), designated the sensor component. The size of the PMUT device is 4 mm × 4 mm, the packaged size is 6 mm × 8 mm, and the size after integration with a microphone is 15 mm × 17 mm. The microphone measures 3.8 mm × 3.3 mm. Compared with the previous device placed on the throat [41], the system in this work demonstrated improved comfort during testing. The package material in this work is JA-2S polyurethane rubber. This material has an acoustic impedance similar to that of human skin, meeting the requirements of package materials for skin patch transducers.

The sensor component is attached to the zygomatic arch using an earphone mold to collect sound by bone conduction or air conduction; the sound is then amplified into an analog voltage [42]. The obtained voltage signals are transferred through a flexible printed circuit (FPC) to an analog-to-digital converter (ADC) for digitization and then processed by a microcontroller unit (MCU). The processed digital signals are transmitted in real time to a mobile application via Bluetooth. The processed digital signals can also be converted back to analog form by a digital-to-analog converter (DAC) and transmitted to the speaker via a USB-C interface.

As the key speech pickup sensor in the bone conduction microphone system, the PMUT is designed with a honeycomb structure [43], as illustrated in Figure 3a. Figure 3b illustrates the operating principle of the PMUT as a bone conduction microphone. The vibration of the bone induces a charge on the surface of the PMUT’s piezoelectric layer, which generates a simulated signal representing speech captured through bone conduction.

Figure 3c illustrates the measured amplitude–frequency response of the reported PMUT under 1 V excitation. The resonant frequency of PMUT is higher than the human voice acquisition frequency range, allowing it to maintain a stable amplitude–frequency response within the human voice acquisition frequency range, thereby ensuring that the collected sound signals are not distorted.

Figure 3d illustrates the measured sensitivity of the reported PMUT (with 60 dB amplification) alongside the typical sensitivity of the commercial ACM within the frequency range necessary for human sound detection. The results indicate that the response of the PMUT remains relatively flat between 20 Hz and 10 kHz. Under a circuit amplification of 60 dB, its sensitivity is −34.5 dB (re: 1 V/Pa). Consequently, the sound signals collected by the BCM and ACM both maintain true proportions without being partially amplified. To match the greater intensity of BC vibrations compared to AC sound waves, a dynamic amplification of approximately 20 dB is added into the circuit for the ACM. It should be noted that the wearing position and method significantly impact the quality of the acquired signals, thereby affecting speech enhancement performance.

## 3. Speech Capture and Visualization

In this section, speech recordings of two different speakers were made using the reported microphone system in various noise environments. Subsequently, the temporal and frequency spectral characteristics of these speech signals, along with their spectrograms, were presented and compared.

Temporal and frequency domain analysis are two important methods for speech analysis; each hass its limitations. Temporal analysis does not provide an intuitive understanding of the frequency characteristics of speech signals, while frequency domain analysis fails to capture the temporal variations of speech signals. The spectrogram combines the advantages of both analyses, clearly displaying the changes in the speech spectrum over time. In a spectrogram, the horizontal axis represents time, and the vertical axis represents frequency, with the intensity of a given frequency component at a specific moment indicated by varying shades of color. The amplitude, fundamental frequency, harmonic frequencies and envelope of the signal correspond to the three essential elements of sound: loudness, pitch, and timbre.

Firstly, the measured voltages of the spoken, BC-captured phrase “Wu Han Da Xue” (“Wuhan University” in Chinese) were collected by the reported PMUT-based system. Their temporal and frequency spectral characteristics are illustrated in Figure 4a,b. The spectrogram are presented in Figure 4c. From the frequency spectrum, the fundamental frequency of the speaker’s voice is 110 Hz, while harmonic frequencies above 1100 Hz are nearly obscured by noise.

Subsequently, the reported PMUT-based system and commercial ACM were employed to capture the speech of different speakers in various environments with different noise levels. The spectrograms of the spoken phrase “Wu Han Da Xue” collected by these two methods are illustrated in Figure 5.

The comparison of the spectrograms of the two speakers clearly indicates that the pitch of Speaker 2’s voice is higher than that of Speaker 1. It can also be observed that the amplitude, fundamental frequency, and harmonics of each character vary significantly. These distinguishing features provide a clear representation of how speech recognition differentiates between texts or speakers. Spectrograms of speech in various noise environments indicate that under quiet conditions, the SNR of the air conduction microphone is slightly superior to that of the PMUT-based bone conduction microphone. However, in noisy environments, the SNR of the air conduction microphone significantly decreases. In contrast, the PMUT-based bone conduction microphone maintains a high SNR across various noise intensities. Table 1 presents the amplitude ratios of BC and AC speech signals to noise measured in environments with different noise intensity.

## 4. Speech Recognition Accuracy Enhancement

In this section, we investigate speech enhancement, using the AC- and BC-captured speech to improve speech recognition accuracy in noisy environments. An overview of the enhanced speech recognition system is shown in Figure 6. The acoustic model training and online decoding in this work are based on the open-source project Kaldi [44]. The speech enhancement in this work is based on PyTorch 2.4.1.

The BC and AC speech signals are initially fused and denoised using the speech enhancement model. Subsequently, they are subjected to online decoding with the acoustic model trained by the speech recognition module, ultimately yielding the recognition results. The online decoding module analyzes the acoustic features of speech using a trained acoustic model, recognizing the corresponding text and outputting it in real time.

In the acoustic model training module, the open-source speech corpus AISHELL-1 [45] was subjected to MFCC feature extraction, and an acoustic model was trained using the features based on the DNN-HMM model. The lexicon and language model required for decoding were generated by manual input and Kaldi’s inbuilt scripts. In the speech enhancement module, the features of datasets are extracted and then input into the AFF [39] and DCCRN [40] modules, resulting in the enhanced speech. In the online decoding module, the acoustic model, dictionary, and language model within the speech recognition module are utilized to analyze the acoustic features of the speech, recognizing the corresponding text and outputting it in real time. Character error rate (CER) is one of the commonly used metrics for evaluating the accuracy of speech recognition [46,47], with the following calculation formula:(1)CER=(S+D+I)/N
where *S* denotes the number of characters replaced in error, *D* denotes the number of characters deleted in error, *I* denotes the number of characters inserted in error, and *N* denotes the total number of characters in the reference text. A lower value of CER indicates better performance of the speech recognition system.

### 4.1. Speech Enhancement Model

To fully exploit the respective acoustic features of AC and BC speech signals, an attentional feature fusion approach is employed. Together with the deep complex convolutional recurrent network, these components form the speech enhancement model in this work. Figure 7a,b, respectively, illustrate the networks of AFF and DCCRN [39,40].

Firstly, the frequency spectra of AC and BC speech signals, which contain their acoustic features, are extracted using the short-time Fourier transform (STFT) and subjected to dimensionality transformation. The processed spectrograms are fed into the AFF module. The AFF module generates a fused frequency spectrum, which is then input into the DCCRN. The DCCRN trains the fused spectrum to approximate clean speech and ultimately outputs the enhanced speech.

### 4.2. Dataset and Setup

Table 2 lists the datasets used for speech enhancement. The original clean dataset used for the speech enhancement model consists of a small dataset containing 40 speech recorded with the reported microphone system. It includes AC speech signals recorded in a quiet environment (AW dataset) and BC speech signals recorded simultaneously with the AC signals using the reported bone conduction microphone system (BW dataset). The noise was sourced from Tsinghua University’s THCH30 dataset [48] (NW dataset), which includes three types of noise, comprising cafe noise, car noise, and white noise. Of these, cafe and car noise are environmental noise.

During the training and testing of the speech enhancement model, Input 1 consists of AW–NW mixtures (ANW dataset), generated by combining the AW and NW datasets at three typical noise levels: −5 dB, 0 dB, and 5 dB. Input 2 consists of the corresponding speech from the BW dataset. The ratio of the training dataset to the test dataset is 4:1.

The loss function for speech enhancement model training is SI-SNR, which is widely used as an evaluation metric. SI-SNR is defined as(2)$$\left\{\begin{array}{c} S=\left(<\tilde{s},s>\cdot s\right)/{∥s∥}_{2}^{2}\\ N=\tilde{s}-S\\ \mathrm{SI-SNR}=10\cdot \log_{10}\left(\frac{{∥S∥}_{2}^{2}}{{∥N∥}_{2}^{2}}\right) \end{array}\right.$$
where s and $\tilde{s}$ represent the clean and estimated time-domain waveforms, respectively. <,> denotes the dot product between two vectors, and ${∥\;∥}_{2}^{2}$ represents the Euclidean norm.

Each of the speech inputs has a sampling rate of 16 kHz, 16-bit depth, and a duration of 5.25 s. Therefore, the number of data points per speech input is 84,000. The frame duration is set to 25 ms, resulting in a tensor dimension of 400 × 210. The hop length is one-sixth of the frame length based on the number of convolutional layers, and the FFT (fast Fourier transform) length is 512. The learning rate is set to 0.001.

Finally, the enhanced speech, AW dataset, ANW dataset, and BW dataset are decoded using the online speech recognition system to obtain their recognition structures and CERs. The acoustic scale of decoding is set to one.

### 4.3. Results and Discussion

Firstly, hybrid training on all noisy speech types is conducted. The CERs across different datasets are compared to evaluate the effectiveness of BC speech in improving speech recognition in noisy environments. Figure 8a illustrates the CERs of different datasets. For the validation dataset of acoustic model training, the CER is 7.4%. For the datasets in speech enhancement module, the CER for the clean AC speech is 7.9%, which is close to the validation dataset’s CER, indicating that the recorded clean speech is of good quality. The CERs obtained from BW dataset decoding are both over 90%. This indicates that the absence of high-frequency features in BC speech prevents it from independently performing speech recognition tasks. After adding noise with levels of −5 dB, 0 dB, and 5 dB to the clean speech, the decoding CERs are 69.1%, 63%, and 50.5%, respectively. The CERs for the enhanced speech are 50%, 48.4%, and 42.3%, respectively, representing reductions of 19.1%, 14.6%, and 8.2% compared to the original noisy datasets. This demonstrates that the speech enhancement method based on the fusion of BC and AC speech effectively improves speech recognition accuracy in noisy environments.

To comprehensively evaluate the speech enhancement performance, perceptual evaluation of speech quality (PESQ) and short-time objective intelligibility (STOI) metrics were incorporated into the hybrid training. As shown in Figure 8b,c, compared with the noisy speech, the PESQ and STOI of the enhanced speech exhibit significant improvements at all noise levels. Specifically, under the low SNR condition of −5 dB, the PESQ and STOI of the enhanced speech are improved by 1.26 and 14.4%, respectively, compared to those of the noisy speech. As the SNR increases, the gains of speech enhancement via BC and AC fusion gradually decrease. This is because under high-SNR conditions, the noise interference is minimal, weakening the advantage that BC speech is less sensitive to noise. Under SNR conditions of 0 dB and 5 dB, the PESQ of the enhanced speech is improved by 1.15 and 0.9, respectively, and the STOI is improved by 10.9% and 7.1% respectively, compared with the noisy speech.

However, the CER of over 40% in speech recognition for the enhanced speech remains significantly high. Comparing the CER of enhanced speech across different types of noise, the CER for speech with added white noise (90.1%) is significantly higher than that for other noise types (35.1%). This is because white noise covers many of the acoustic features of speech, resulting in the discarding of these acoustic features during the training process. As a result, although the PESQ and STOI of the enhanced speech improve, accurate speech recognition remains unachievable.

Figure 9 illustrates a comparison of the Mel spectra of noisy and enhanced speech in environmental and white noise. It is evident that the environmental noise suppression effect is superior. In contrast, white noise causes severe contamination across the entire frequency band of speech, which leads to difficulties in recovering some acoustic features.

White noise was then removed from the types of added noise, and separate training of noisy speech without white noise is conducted. The resulting CERs are shown in Figure 10a. The decoding CERs of noisy speech with −5 dB, 0 dB, and 5 dB were 53.6%, 45.2%, and 25.9%, respectively. The CERs for the enhanced speech were 26.9%, 13.6%, and 8.1%, respectively, representing reductions of 26.7%, 31.6%, and 17.8% compared to the original noisy datasets. The enhanced speech at 0 dB and 5 dB noise levels has achieved speech recognition capabilities comparable to that of clean speech.

Subsequently, we conducted a separate speech enhancement for the noisy speech only with white noise. The speech enhancement reduced the CER of the noisy speech from over 99% to 84.5%. Next, by decreasing the white noise intensity, the CER of the original noisy speech is 80.2% at a noise level of 10 dB, while the CER of the enhanced speech is reduced to 15.4%, as illustrated in Figure 10b. It indicates that the speech enhancement method is effective in suppressing non-extreme white noise.

The problem of weak robustness to full-band white noise is preliminarily alleviated by reducing the white noise intensity to 10 dB, though the problem has not been completely resolved. In subsequent study, targeting the characteristics of white noise, we will adopt a method combining DSP algorithms with deep learning and optimize the speech enhancement model, so as to achieve better versatility and performance.

Furthermore, the dataset size of 20 speech samples is relatively small. We used 20 samples to preliminarily verify the feasibility of the BC and AC speech fusion enhancement method and the application potential of PMUT. After expanding the samples to 40, the speech enhancement effect has slightly improved, but the overall trend remains unchanged. In subsequent experiments, we will further expand the size of the original speech dataset and include different speakers to improve the model’s generalization ability, thereby forming a more comprehensive research framework.

## 5. Conclusions

In this work, a customized bone conduction microphone system based on a PMUT is developed to capture real-time BC speech signals, and an ACM is integrated for simultaneous capture of AC speech signals. The PMUT-based BCM system achieves a simpler and more robust device for effective continuous capture of BC speech. The BC-captured speech achieves a signal-to-noise amplitude ratio over five times greater than AC-captured speech in an environment with 68 dB noise. Subsequently, an enhancement method based on the fusion of BC and AC speech signals was used to improve the accuracy of speech recognition in noisy environments, demonstrating the application potential of this BCM system.

For hybrid training with various noise types, the CERs of enhanced speech are 50%, 48.4% and 42.3% at −5 dB, 0 dB, and 5 dB noise levels, respectively. Compared with the original noisy dataset, the CERs are reduced by 19.1%, 14.6% and 8.2%, respectively. Following the removal of white noise, the CERs of the enhanced speech improve further, recording values of 26.9%, 13.6% and 8.1% at −5 dB, 0 dB, and 5 dB noise levels, respectively. These CERs represent reductions of 26.7%, 31.6% and 17.8% compared with the original noisy dataset, respectively. Additionally, in separate training at a 10 dB white noise level, the enhanced speech achieved a CER of 15.4%. Collectively, the results indicate that the speech enhancement strategy based on the feature fusion of BC and AC speech in this work effectively suppresses environmental noise and non-extreme white noise, approaching the speech recognition accuracy of clean speech.

This work focuses on efficiently capturing BC speech using PMUT and using the acoustic feature advantages of BC- and AC-captured speech for speech enhancement, without improving the network structure of the speech enhancement used. Moreover, the speech datasets in this work consisted of unprocessed raw speech, highlighting the practicality of this system for applications. However, this also brings disadvantages in terms of SNR and recognition accuracy. In future study, we plan to increase the size of the speech dataset, preprocess the speech data, and optimize the speech enhancement model for white noise to improve the generalization ability of the model and achieve better speech recognition accuracy. Additionally, in terms of hardware, we will utilize ASIC and integrate the system with AI chips to achieve a system with higher integration and lower power consumption.

## Figures and Tables

**Figure 1 micromachines-16-00613-f001:**
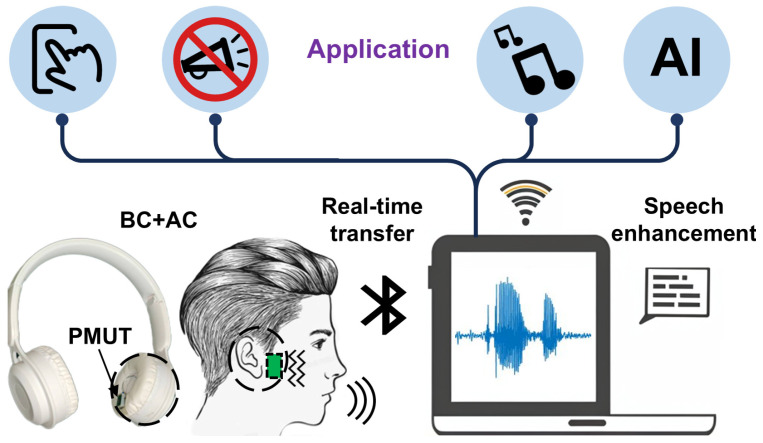
Overview of the system application developed in this work. BC: bone conduction; AC: air conduction.

**Figure 2 micromachines-16-00613-f002:**
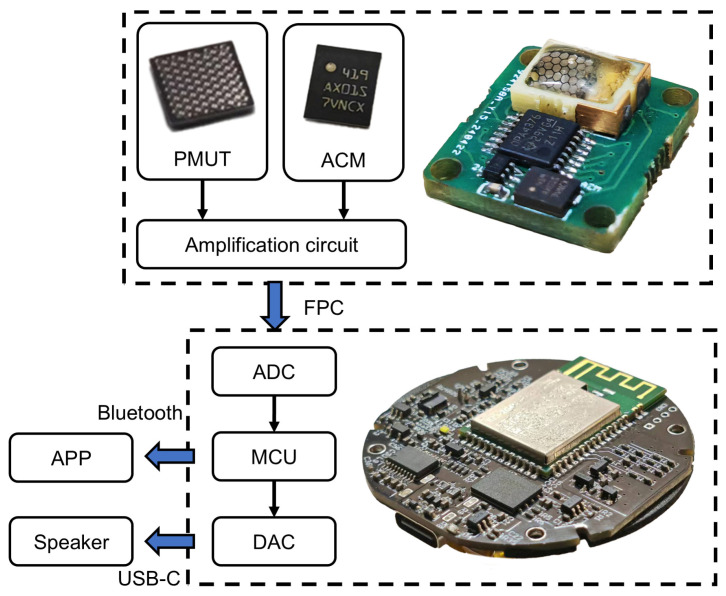
Schematic diagram of the reported PMUT-based bone conduction microphone system.

**Figure 3 micromachines-16-00613-f003:**
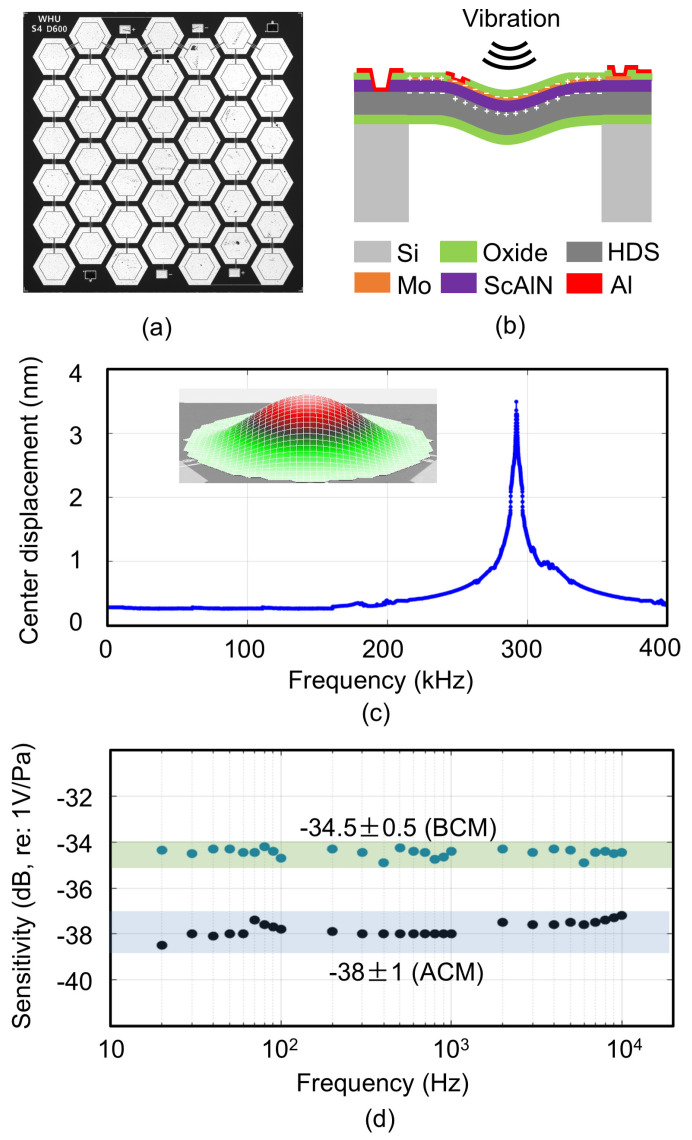
(**a**) Optical microscope photograph of a honeycomb PMUT. (**b**) Operating principle of the PMUT as a bone conduction microphone. HDS: highly doped silicon. (**c**) Measured amplitude–frequency response of the reported PMUT under 1 V excitation. (**d**) Measured sensitivity of the reported PMUT within the frequency range required for human speech pickup and frequency response of the ACM according to the data sheet.

**Figure 4 micromachines-16-00613-f004:**
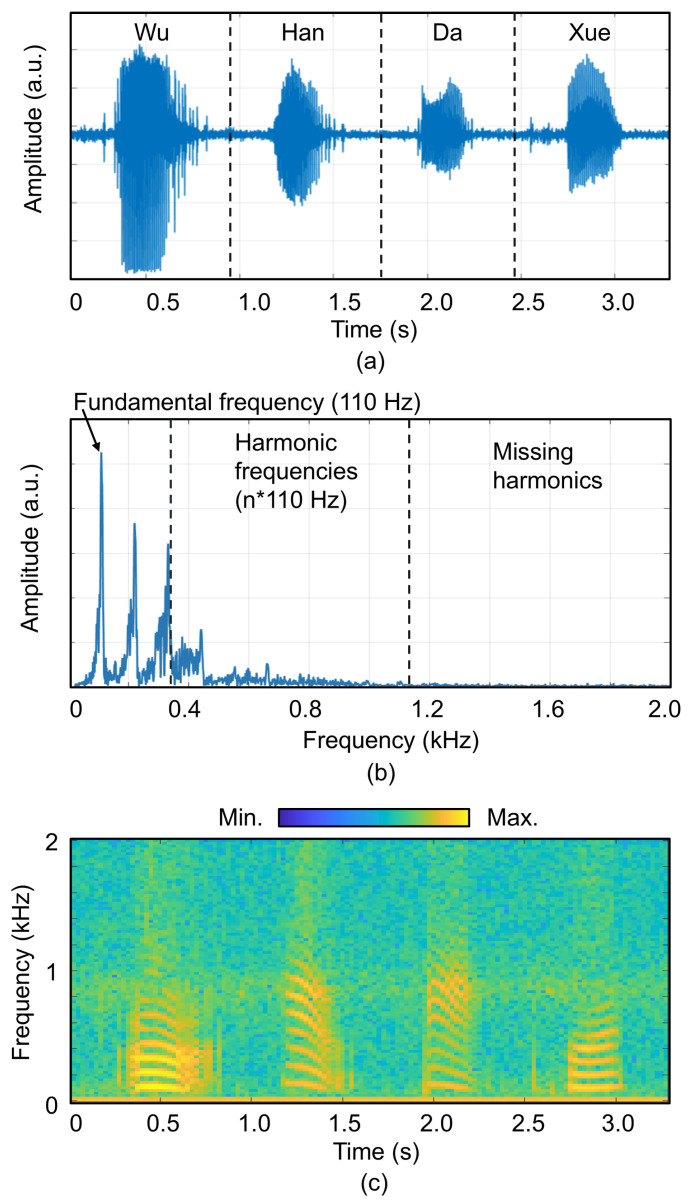
(**a**) Time domain, (**b**) frequency domain, and (**c**) spectrogram of the sound signal “Wu Han Da Xue” (“Wuhan University” in Chinese) collected by the reported PMUT-based system.

**Figure 5 micromachines-16-00613-f005:**
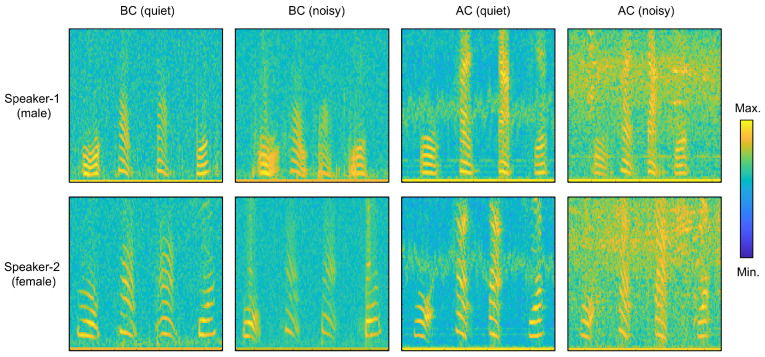
Spectrograms of the speech “Wu Han Da Xue” collected from 2 speakers using the reported BCM and ACM in different environments.

**Figure 6 micromachines-16-00613-f006:**
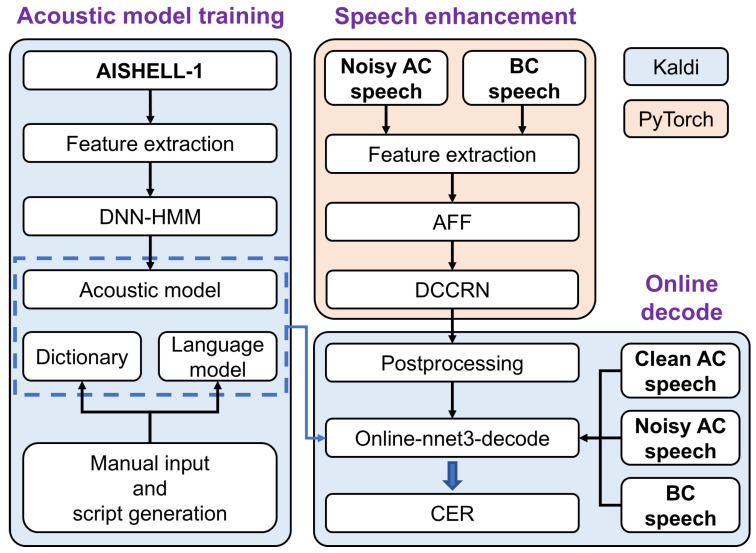
Speech recognition framework in this work. The entire work encompasses the acoustic model training, speech enhancement model, and online decoding system.

**Figure 7 micromachines-16-00613-f007:**
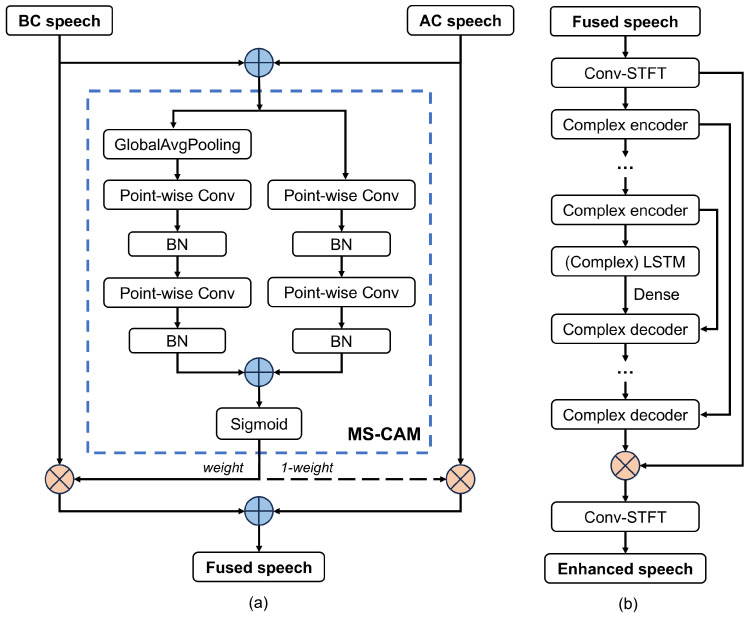
Networks of (**a**) AFF [39] and (**b**) DCCRN [40]. The first Conv-STFT is placed before the training of the AFF-DCCRN.

**Figure 8 micromachines-16-00613-f008:**
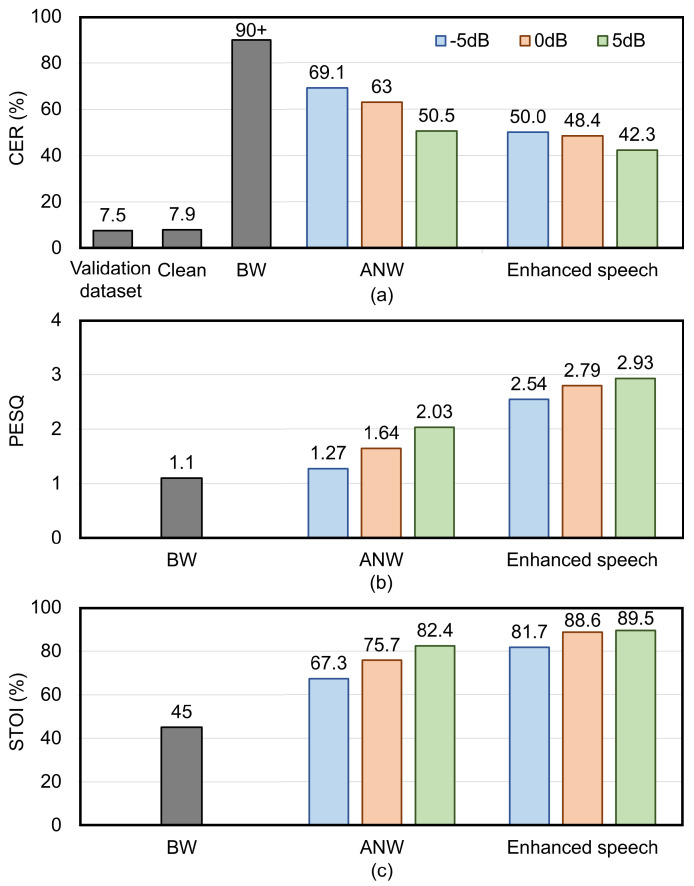
Comparison of (**a**) CERs, (**b**) PESQs, and (**c**) STOIs for different datasets during hybrid training, including the validation dataset of the acoustic model, the AW and BW datasets, and the ANW and enhanced speech datasets with various noise levels.

**Figure 9 micromachines-16-00613-f009:**
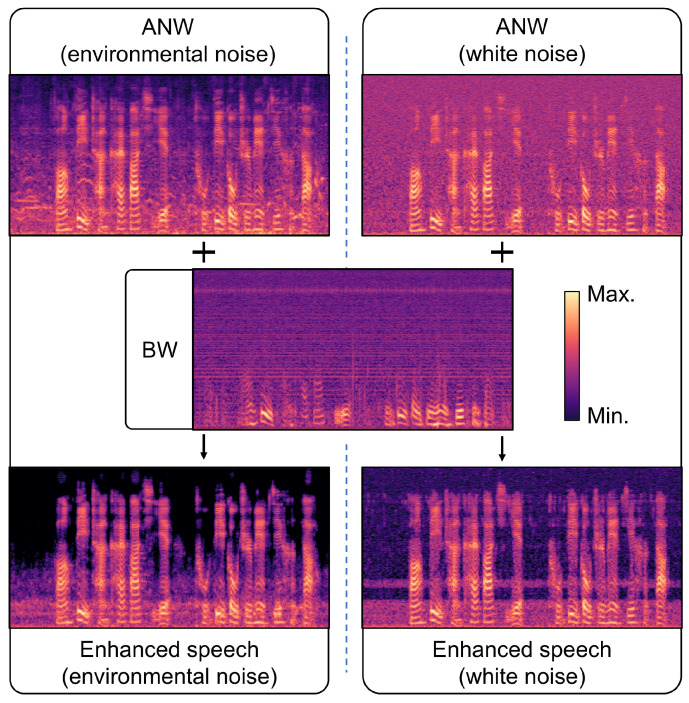
Comparison of the Mel spectra of noisy and enhanced speech in environmental and white noise.

**Figure 10 micromachines-16-00613-f010:**
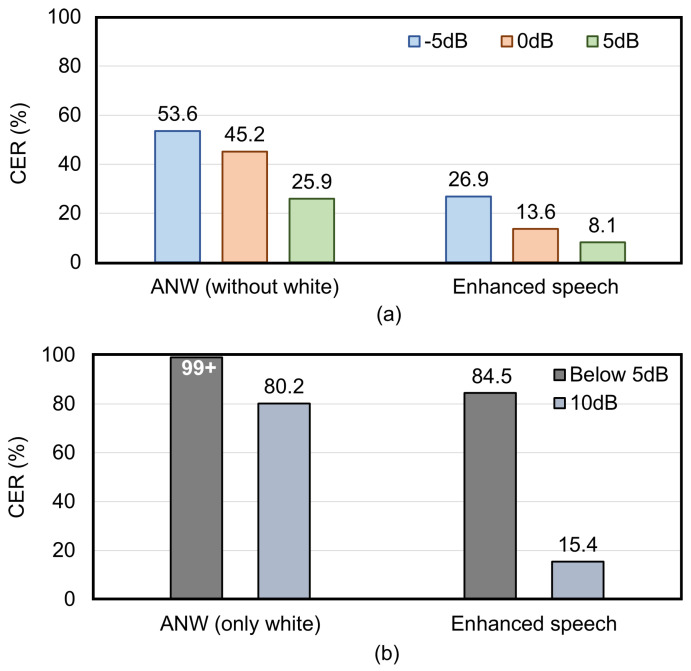
Comparison of CERs for different datasets during separate training. (**a**) includes the ANW and enhanced speech datasets that exclude white noise. (**b**) includes the ANW and enhanced speech datasets with only white noise (“Below 5 dB” represents the average value of −5 dB, 0 dB, and 5 dB).

**Table 1 micromachines-16-00613-t001:** Measured amplitude ratio of speech signals to noise for BC- and AC-captured speech in various environments with different noise levels.

Noise Environment	AC	BC
Quiet (∼40 dB)	150	70
Noisy (∼60 dB)	20	60
Noisy (∼68 dB)	10	55

**Table 2 micromachines-16-00613-t002:** Information On the datasets used for speech enhancement.

Dataset	Speech Source
AW	Captured clean AC speech
BW	Captured BC speech
NW	Noise from THCH30 [48]
ANW	Noisy AC speech obtained by mixing AW and NW datasets

## Data Availability

The raw data supporting the conclusions of this article will be made available by the authors on request.

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
