# Peer review of "A Piezoelectric Micromachined Ultrasonic Transducer-Based Bone Conduction Microphone System for Enhancing Speech Recognition Accuracy"

_micromachines, 2025, doi:10.3390/mi16060613_

Round 1

Reviewer 1 Report

Comments and Suggestions for Authors

This manuscript proposes a bone conduction microphone system based on PMUT, combined with a multimodal speech enhancement algorithm, aiming to solve the problem of the decline in speech recognition accuracy in a noisy environment. This manuscript designed a PMUT with a honeycomb structure, achieving a simplified bone conduction microphone. The manuscript innovatively proposed a multimodal speech enhancement framework based on AFF and DCCRN, improving the performance of speech recognition in noisy environments. In terms of experimental verification, through comparative tests of multiple scenarios and multiple noise types, the manuscript confirmed that the SNR of BC speech in a 68 dB noise environment was more than five times that of AC speech, demonstrating strong practical potential.

  1. In terms of writing and presentation, the manuscript has a clear structure, detailed charts, and strong logical coherence. However, attention should be paid to the uniformity of terms. For example, when abbreviations (such as AF-DCCRN) appear for the first time, the full names should be marked.

  1. The scale of the experimental dataset containing 20 self-collected voices is relatively small, limiting the generalization ability of the model. The authors may increase the numbers.

  1. Although the author alleviated the problem of weak robustness to full-band white noise by reducing the intensity of white noise (10 dB), targeted solutions such as frequency band selective suppression or model structure optimization were not explored. The authors may add an explanation in the results analysis and discussion part.

  1. The authors may supplement the discussion on the hardware cost and power consumption of PMUT to enhance the discussion on the application value of the research.

  1. In line 164, the authors can provide some supporting literature for CER to explain why CER is a commonly used indicator.

  1. The authors should clarify which part is PMUT in Figure 1.

Reviewer 2 Report

Comments and Suggestions for Authors

This work aims to develop a customized bone conduction microphone (BCM) system based on a piezoelectric micromechanical ultrasound transducer. This work also introduces a simpler and more powerful device for efficient continuous capture of BC speech. While this work is well written and thorough, I have the following comments for further improvement:

1. In Figure 3(b), the full name “HDS” is not mentioned and should be given in the figure.

2. The authors should give the dimensions of the PMUT device, the reported PMUT-based bone conduction microphone system, and the wearing comfort in the tests.

3. The packaging material of the PMUT in this article should also be provided to evaluate its biocompatibility, if different from that in Reference 39.

4. Since the PMUT must be used through close contact, it should be clarified whether the effect of speech enhancement is sensitive to the wearing position and wearing style.

5. Since the paper reports that the SNR of air-conduction microphones is slightly better than that of PMUT-based bone-conduction microphones under quiet conditions, the author may give some explanation.

6. Speech enhancement was performed only for noisy speech with and without white noise, but the author may also consider speech enhancement with café noise, car noise, and attenuated white noise to further determine the effectiveness of low white noise for practical applications.

Reviewer 3 Report

Comments and Suggestions for Authors

This is a very interesting work in such a practically important area as voice recognition in a noisy environment. The work consists of three parts: a description of the innovative Bone Conduction Microphone System, a description of the performance of the resulting hybrid system for recording human speech, and improving the quality of recorded speech using modern neural networks. The work is done at a high scientific level and I have only a few small comments about the pictures.

1. In Fig. 8, the designation Dev (Validation dataset) should be clearly deciphered.
2. In Fig. 10(b), the designation "blow 5dB" is unclear.

Reviewer 4 Report

Comments and Suggestions for Authors

Overall, I liked the work submitted for review. In this work, a customized bone conduction microphone (BCM) system based on piezoelectric micromachined ultrasonic transducer is developed to capture the real-time bone conduction (BC) speech, while a commercial ACM is integrated for simultaneous capture of air conduction (AC) speech. However, I would suggest that the author revise the manuscript taking into account my comments, which I outline below:

1. In a review of competing sound recording methods, it would be appropriate to mention fiber-optic sensors. These days, they show amazing results in voice detection and recognition, even in the case of a distributed operating mode:

http://dx.doi.org/10.1109/ECOC48923.2020.9333283

https://doi.org/10.3390/s24072281

2. In my opinion, the work does not pay enough attention to the physical parameters of digitization. Thus, knowing the characteristic amplitude frequency response of the presented microphone, it would be possible to study various sampling frequencies and their effect on the quality of recognition.

3. By the way, it would be appropriate to attach the microphone's amplitude-frequency characteristic to the manuscript.

4. It is known that preliminary data processing before recognition significantly improves the quality of this process. Why didn't the authors pay attention to such methods as wavelet filtering, as well as empirical and variational mode decomposition?

5. More data on word sets and formal criteria used to evaluate speech recognition quality should be presented. Harvard sentences and various metrics, such as Levenshtein distance, are usually used for this.

6. Some figures are located directly at the end of sections, I would suggest moving them up, right after the first mention in the text.

Round 2

Reviewer 1 Report

Comments and Suggestions for Authors

No 

Author Response

Thank you very much.